# Personalized music for cognitive and psychological symptom management during mechanical ventilation in critical care: A qualitative analysis

**Rebecca Menza** [1,2] *, **Jill Howie-Esquivel** [3], **Tasce Bongiovanni** [4], **Julin Tang** [5], **Julene K. Johnson** [2], **Heather Leutwyler** [3]

1 Department of Trauma and Critical Care Surgery, Zuckerberg San Francisco General Hospital, San Francisco, California, United States of America, 2 Institute for Health & Aging, University of California San Francisco, San Francisco, California, United States of America, 3 Department of Physiological Nursing, University of California San Francisco, San Francisco, California, United States of America, 4 Department of Surgery, University of California San Francisco, San Francisco, California, United States of America, 5 Department of Anesthesia, University of California San Francisco, San Francisco, California, United States of America

* Rebecca.menza@ucsf.edu

**Data Availability Statement:** All relevant data are contained within the manuscript and/or Supporting files. The raw data are transcribed interviews and

## Abstract

### Introduction

Patients experience high symptom burden during critical care hospitalization and mechanical ventilation. Medications are of limited effectiveness and are associated with increased morbidity such as delirium and long-term cognitive and psychological impairments. Music-based interventions have been used for pain and anxiety management in critical care but remain understudied in terms of music selection and range of symptoms. This study aimed to describe the ways in which a diverse sample of critically ill adults used personalized music listening and their perceptions of the effects of music listening on symptom experience after critical injury.

### Methods

Semi-structured interviews (N = 14) of adult patients, families and friends who were provided with personalized music in an urban, academic, neurotrauma intensive care unit were collected and analyzed with grounded theory methodology. Open coding of transcripts, field notes and memos was performed using Atlas.ti.9.1. Recruitment and data collection were deemed complete once thematic saturation was achieved.

### Results

We identified 6 uses of personalized music listening in critical care: 1) Restoring consciousness; 2) Maintaining cognition; 3) Humanizing the hospital experience; 4) Providing a source of connection; 5) Improving psychological wellbeing; and 6) Resolving the problems of silence. Patients used music to address psychological experiences of loneliness, fear,

coding notes, which are stored securely with the corresponding author. Data cannot be shared publicly because the small number of deeply personal interviews contain highly sensitive material, collected from vulnerable people, during a specific period of time posing risks that participants may be able to be identified through indirect means. Data are available from the University of California San Francisco Institutional Data Access / Ethics Committee (contact via email LibraryDSOS@ucsf.edu and or irb@ucsf.edu) for researchers who meet the criteria for access to confidential data.

**Funding:** The author(s) received no specific funding for this work.

**Competing interests:** The authors have declared that no competing interests exist.

confusion, and loss of control. Personalized music helped patients maintain their identity and process their trauma. Additional benefits of music included experiencing pleasure, hope, resilience, and feelings of normalcy. Patients disliked being sedated and used music to wake up. Findings also highlighted the problem of the lack of meaningful stimulation in critical care.

## Conclusion

Critically injured adults used personalized music to achieve psychological and cognitive homeostasis during critical care hospitalization. These results can inform future studies designed to explore the use of music-based interventions to prevent and treat the cognitive and emotional morbidity of critical care.

## Introduction

Critically ill and injured adults have a significant symptom burden, often reporting symptoms of pain, anxiety, restlessness, dyspnea, confusion, and sleep disorders [1, 2]. Mechanical ventilation contributes to communication deficits while also being an independent risk factor for each of these distressing symptoms [3]. Medications to manage these symptoms in ICU have limited effectiveness and are associated with negative side effects including increased risk of delirium [4–6]. Patients in ICU also report a series of psychological symptoms not amenable to pharmacologic interventions, including loneliness, fear, and depressive feelings [2, 7–9]. Undermanaged psychological symptoms and delirium contribute to the development of long-term psychological and cognitive symptoms including anxiety, depression, post-traumatic stress, cognitive impairment, functional disability and decreased quality of life, a variety of symptoms often referred to as post-intensive care syndrome [10, 11]. Increased awareness of the potential medication side effects alongside the imperative to manage morbidity from distressing symptoms has led to increased interest in nonpharmacologic interventions for symptom management in ICU, such as music-based interventions.

When music is used in the clinical setting to achieve a health-related goal such as symptom management, it is called a music-based intervention (MBI) [12]. MBIs may be delivered by a credentialled music therapist with specialized training (music therapy) or by a health care provider (music medicine) [13, 14]. Listening to recorded music is a widely requested type of MBI in critical care units [15–17]. The use of MBIs to manage pain and anxiety appears in several summary recommendations as accessible, acceptable, low risk interventions to promote safe and effective analgesia and anxiolysis in critical care [12, 18, 19].

Prior studies examining the use of MBIs in critical care, and among MV adults, suggest that music listening may be an effective adjunct for the management of pain and anxiety [20, 21]. However, most clinical trials of MBIs in critical care suffer important limitations threatening their generalizability. First, despite recommendations to use patient preferred music for symptom management in critical care [22], most studies of MBI in MV adults use music selected by investigators or limit patient choice [23]. This practice may contravene the well described relationship between preference and familiarity and the psychobiological effects of music listening [24–27]. Lack of participant socio-cultural diversity is another important limitation in North American studies of MBI in MV adults. This is important because the cognitive, processing, affective nature of, and response to music depends on an individual's social context, cultural

values and prior experiences [28]. Given that music preference and familiarity vary with socio-cultural background [28–30], inclusion of a more culturally diverse group of participants may also result in a wider range of 'preferred' or 'familiar' music. Finally, the majority of clinical trials examining the effects of MBIs in critical care only explore outcomes of pain and anxiety. These narrowly focused studies may overlook upstream causes of pain and anxiety such as impaired communication and sleep disturbances, or important symptom experiences such as loneliness and fear.

Soliciting the views of patients who listened to music during their critical care hospitalizations through qualitative exploration may reveal a wider range of experiences of music listening. Agnostic exploration of the use of music listening in ICU may also help explain the mechanism of action of MBIs more generally. Therefore, the purpose of this study is to describe patients' perceptions of the effects of listening to self-selected music on symptom experience during MV after critical injury.

## Theoretical framework

The use of music as a therapeutic intervention is supported by two important theories, each based in the philosophical principles of holism: Antonio Damasio's Theory of Emotion, Feeling and Core Consciousness, [26] and Florence Nightingale's Theory of Nursing (and the Environment) [31]. Holism emphasizes the connection between the body and the mind and acknowledges the inseparable relationship between the two [32]. Damasio posits that homeostasis and, ultimately, survival, are dependent on the relationships between physical feelings in the body proper, and emotions developed in the brain and consciousness. According to Damasio, memories of physical and emotional perceptions form a complex network, or neural map, of experiences unique to each individual. Music, a strong auditory stimulus, is processed in parts of the brain that are responsible for emotional regulation and homeostasis. Nightingale also identified music as a powerful sensory stimulus enabling nurses to alter the environment surrounding a patient, while providing individualized care. According to Nightingale, wellness and recovery can be achieved through experiences such as music listening which promote harmony of mind, body, and spirit. As such, these theoretical frameworks are well suited to inform a rich exploration of the ways critically ill and injured adults use music during their hospitalizations.

## Methods

### Setting, participants and study design

We used Grounded Theory methods to answer the research question [33]. This study was conducted at an urban, academic level-1 trauma center and safety net hospital. Inclusion criteria were: age 18 years old or older; current or recent patient hospitalized in the trauma and neurological ICUs or family member; having experienced MV during the hospitalization; having listened to recorded music in the ICU as a patient or having played recorded music for patients in the ICU as a family member. Participants for this convenience sample were identified and recruited in 2 ways: 1) through census review by one of the investigators (RM), who is also an advanced practice provider in this ICU; or 2) through referral from critical care clinicians who: played recorded music for their patients, facilitated music listening for patients through patient or family request, or observed their patients listening to recorded music during their hospitalization. Purposive sampling was used to ensure a socially, culturally, ethnically, linguistically, and economically diverse sample of respondents and was ongoing until thematic saturation was achieved. Institutional Review Board approval was obtained from the Committee for Human Research at the University of California, San Francisco (IRB#18–26580). Informed

consent was obtained for all interviews. This study adhered to the COREQ guidelines for reporting qualitative research [34].

## Recorded music listening

Recorded music was sourced and played by patients, families and staff using a variety of means throughout the study period including: through web-based subscription services (e.g., Spotify), streamed radio broadcasts, or from personal devices brought to the bedside by patients and their families. Music was provided to unconscious patients through portable speakers placed near their hospital beds. Conscious patients used portable speakers or headphones depending on their preference. In general, families made music selections for unconscious patients. Once they regained consciousness, patients made their own selections. There were no restrictions placed on the genre, tempo, mood, rhythm, or lyrics of the music selections. Patients listened to music at any time of day and for any length of time.

## Data collection and analysis

Recruitment and data collection took place from August 2020 to November 2021. The primary source of data were semi-structured interviews conducted by the principal researcher (RM). An interview guide was developed for the study (S1 File) and was adapted throughout the study period according to participant responses. Interviews were recorded and transcribed verbatim (RM) except for 2, where detailed notes were taken. Interview transcripts were checked against recordings to ensure accuracy. Summaries were shared with participants at the close of each interview as a form of member checking and to clarify concepts. In addition to the interviews, field notes were collected by the principal researcher (RM) throughout the study period including quotes and observations made by additional informants including clinicians, patients, and families.

All interview transcripts, summaries and field notes were entered into Atlas.ti 9.1 [35]. Data collection and analysis, including memo writing, and constant comparison was concurrent throughout the study period, consistent with Grounded Theory methodology [33]. Word by word and line by line open coding of all transcripts, field notes and memos was performed by a single researcher (RM). Family-member perspectives were coded for their views of the patients' experiences and were used to confirm and illustrate themes and concepts identified in the patient interviews. Open coding was centered on actions (over nouns) to maintain context and rich description. Next, focused codes were created from the open codes and from overall impressions of the data (Table 1). All transcripts were also read by 3 co-investigators (HL, JHE, TB) who validated the focused codes and assisted with coding through discussion of agreed on emerging categories, and relationships between emerging themes. As themes were explored, transcripts and notes were again reviewed by the research team and analyzed for exemplars of emerging themes and divergent cases, staying vigilant to distinct voices and contrary experiences. Individuality was maintained through illustrative quotes and exemplars. The analysis was deemed complete once thematic saturation was reached, and categories and subthemes were organized into a coherent whole.

Throughout this process of reflection, co-investigators wrote memos and used journals to make note of their own perspectives on the stories they heard and the situations they observed in order to remain aware of prior points of view and how these may have changed or influenced coding and interpretations. All data including audio recordings, transcripts, notes, memos, codes, early frameworks, and journal entries was saved. Member-checking with study participants was ongoing throughout the research including during interviews, in order to both sharpen understanding and allow for member corrections. As well, analyses and results

**Table 1. Focused codes.**

| FOCUSED CODES | |
|---|---|
| Advocacy (Care/Respect) | Increasing communication |
| Alternative to medications | Individuality |
| Amnesia | Intrusive thoughts/PTSD |
| Anxiety | Involving Family |
| Being Seen | Joy |
| Boredom/Feeling Stuck | Loneliness/Alone |
| Breathing Difficulties | Making music |
| Calm/Becalming | Mechanistic/Dehumanized/Sterile |
| Cognition | Memories |
| Coma | Mind |
| Comfort/Soothe | Mindset |
| Communicating (when you can't speak) | Mood |
| Community | Music as a visitor |
| Confusion | Noise/Unpleasant sounds |
| Connection | Normal |
| Consciousness | Numbness/Resignation |
| Control | Pain |
| Coping/ inner strength | Peace (Love/Pleasure) |
| Dancing (moving to music) | Positive thinking |
| Delirium | Presence |
| Disconnected/De-situated | Processing emotion (grief/loss) |
| Distraction | Psychological pain (relief) |
| Emotional pain | Relaxation |
| Encouragement | Religion/ Spirituality |
| Enhanced focus | Resilience |
| Environment | Restraint |
| Familiar | Sadness |
| Family History | Separation |
| Fear | Silence |
| Focus | Sleep |
| Gratitude | Social |
| Happiness | Soul |
| Healing | Stimulating senses |
| Homelife | Surviving/Doing Hard Things |
| Hope | Unfamiliar |
| Humanizing Identity | Waking Up |

were shared with key informants, members of the community, and through peer review with critical care clinicians.

## Results

Fourteen semi-structured interviews were conducted with 21 informants during the study period. Interviews were conducted in person (n = 12), over the phone (n = 1) or via video conference (n = 1). Interviews were approximately 40 minutes in length. Interviews were conducted with a combination of patients (n = 7), family members of patients (n = 3), or both (n = 4). Patients were 22–64 years old. Twelve were hospitalized after sustaining critical

polytrauma including 7 who also had a traumatic brain injury (TBI) and 2 patients were admitted after a stroke. Half of the patients were hospitalized after an assault including 6 with gunshot wounds. Four patients identified as Asian (Filipino, Japanese, Uyghur and Chinese), 2 as Black/African American/African, 2 as LatinX (El Salvadoran and Mexican American), and 6 as White/European. Five patients reported having a religious faith (Christian, Buddhist, Muslim and Catholic). All of the patients identified as male. Most family-participants were female (6 of 7). All but one patient spoke English, and for that interview a certified interpreter was used for interpretation to Spanish. Five patients were born and raised outside of North America (China, Western Europe, Northern Africa, Pacific Rim). Half of the patients were employed, and most were housed (n = 11). One patient had formal musical training and 2 others identified as musicians. Thirteen interviews were conducted during the hospitalization, at the bedside in private rooms. One interview was conducted 7 months after admission. In addition to the semi-structured interviews, the primary investigator (RM) documented statements and observations about the use of recorded music in critical care made by other key informants such as patients, families, and staff (S2 File). Quotations presented in this analysis are labeled with anonymized study identification labels; wherein 'P' indicates a patient quote and 'FP' indicates a quote from a patient's friends or family.

## Music selections

Initial music selections were made by family members (n = 7), patients (n = 4) and a combination of family and patients (n = 3). Selections varied widely. Though not formally catalogued, examples of the music selections are presented in Table 2.

## Themes

Overall, patients and families expressed enthusiasm and gratitude for the offer and use of music during their ICU hospitalizations. Six inter-related themes emerged describing patient and family perspectives of listening to recorded music and the effect of personalized music on

**Table 2. Music selections.**

| Participant | Music chosen by | Music Selections |
|---|---|---|
| 1 | Family | Brahms, Hungarian rock of the 70s,Pop Rock (UB40, Beatles); Radio: 'The Bone' Classic Rock |
| 2 | Family | Jesus Adrien Romero [Spanish] Christian Rock/Ballads |
| 3 | Family | Vivaldi 4 Seasons, Soundtrack to Forrest Gump, Traveling Wilburys |
| 5 | Family/Self | Classic Reggae (Bob Marley) Eason Chan |
| 7 | Self | Santana |
| 8 | Self | Corridos Tumbados/Trap, Latinx Pop (Junior H), Reggaeton, [Female] HipHop |
| 9 | Family | Bay Area Hip Hop (Too$hort, E40), Earth Wind and Fire, Miles Davis |
| 10 | Family | Classic Jazz (Mingus, Monk, Peterson, Parker, Coltrane), and Tom Petty |
| 11 | Family/self | Psychedelic /Inde Rock Pop, Synthpop (Tame Impala) |
| 12 | Self | New Wave; Post-punk (The Smiths, The Cure, Echo and The Bunnymen, Crowded House) |
| 13 | Family | 80s and 90s Hip Hop (Tupac, Salt N Pepa) |
| 14 | Family | Latin Jazz (Gabrielle Y Rodrigo) and 'Classic'/"Lounge Jazz (Monk, Coltrane, Peterson, Mingus) |
| 15 | Family/Self | Classic Jazz (Brubeck, Monk, Peterson) and Westen Classical (Mozart) |
| 17 | Self | Classic Jazz (Mingus, Monk, Parker, Coltrane, Peterson) and 'Ethiopian' Music |

**Table 3. Symptoms and experiences affected by personalized music listening in critical care.**

| | |
|---|---|
| Loneliness | Coping |
| Separation | Belonging |
| Fear | Happiness |
| Isolation | Joy |
| Lack of sleep | Hope |
| Anxiety | Resilience |
| Pain | Connection |
| Grief | Mood (Improved) |
| Flashback | Focus |
| Posttraumatic Stress | Pleasure |
| Depression | Normalcy |
| Sadness | Self Determination |
| Dehumanization | Being Seen |
| Intrusive thoughts | Identity |
| Delirium | |
| Confusion | |
| De-situatedness | |
| Sensory deprivation | |
| Boredom | |
| Loss of control | |
| Dependence | |
| Numbness | |

hospital experiences and symptoms (Table 3). Participants described that listening to recorded music: 1) Restored patients to consciousness; 2) Maintained their cognition; 3) Humanized the hospital experience; 4) Provided a source of connection; 5) Improved patients' psychological wellbeing; and 6) Resolved the problems of silence. Each of these 6 themes are distinct albeit some subtlety. Several themes address the same dynamic from slightly different perspectives, e.g. consciousness and cognition.

## Restoring consciousness

Both patients and their family members described the use of music listening as a way to restore patient consciousness after anesthesia, traumatic brain injury, sedation, delirium, and stroke. Music served as a trigger to awaken an otherwise dormant mind.

*. . .it definitely brought me out of that little coma or the little sleep I was in. Like it just shook me and was like 'Hey Buddy'. . .'Now's your time'. . . 'You're up! You're here. You made it!' . . . it was . . . <u>waking</u> me up it was bringing me home. And I felt I was far away. (P7).*

Others felt that music was encouraging them back to awakeness. One patient explained,

*It's like music it's telling you 'Hey, you belong in this world, don't drift away to somewhere else'. . .specially people in coma. Like, 'don't drift away to coma'. . . 'you belong here. You need to wake up. . .eventually. (P5)*

Patients wanted to be conscious. For many, being conscious was a way to reassure people of their survival, and listening to music helped accomplish this. *I remember thinking I really wanted to wake up. . . [music] . . . brought me back. . .music saved my life' (P11)*. This perspective was echoed by many who saw listening to familiar music as a way to '*bring you back*' after coma, to '*stimulate the brain, wake them [sic] up' (P10)*. Patients explained that listening to music prompted the recall of their memories. Memories came '*flooding back in*' and served as a '*trigger*' that helped to '*jumpstart*' patients' consciousness. One patient explained that music

*'. . .sparks some. . .mind. . .creativity? . . . some type of state where you're reminded of what your life was like, or what your life is. . . it. . .made me wake up. . .(P11).*

Family members also chose music with an aim to help restore consciousness through music-related memories. One family member described that she played music associated with significant life events in the hope that the memories would get *'him to come back to us'* (F1). Familiar music was, for many, an accessible sensory lifeline that people could *'grab hold of and use'* to pull themselves out of a *'fog'* into a more conscious, awake and aware state. One man explained,

*. . .it definitely..pulled me out of my . . .submission. . . I just wanted to sleep and be done with it . . . hearing the music it was waking me up because it was familiar. . .something that I know and I love . . . I was in such a flimsy state. . .when I heard that music, I was just like 'I'm here, I'm alive', and I was fighting to get back to being awake and conscious (P7).*

Clinicians and family members also observed more wakeful states and signs of improving consciousness in patients when listening to music, such as smiling, bobbing their heads and tapping their feet. In one instance, a previously non-interactive patient scowled at the staff when they removed the tablet used for playing music, staring at them until they returned the device. One woman described the moment when she saw the first signs of her father's emergence from coma,

*. . . I was . . . recalling how . . . we danced at my wedding and this song played and asking him whether he remembered . . .he was, limited in his communication to just head nodding but.. there was something in his eyes that sparked something . . .with him nodding his head too. . .-like OK he's coming back, he's coming back (FP1)*

## Maintaining cognition

Similar to the use of music listening for the restoration of consciousness, music listening was seen by many as an accesible catalyst with which to initiate and maintain cognitive function, even during emergence from coma. One family described how they hoped to engage their loved one's brain and stimulate thought processing while recovering from a stroke by playing familiar music. *'To hear something that's familiar. . . that will trigger their memories of something as their body is recovering from whatever trauma that they're going through. . . . . ..memories of growing up maybe trigger something in his brain. . .' (FP9)*

Memories triggered by listening to music served as both a starting point, to initiate awakening, and later as a guide by which to traverse from coma and confusion to alertness. One man explained, *'a familiar song can take a person from a place of extreme confusion and unfamiliarity to a safer, kind, happy and ultimately familiar place'.*

The use of sedatives was disorienting for many, leaving patients feeling confused and scared. Some used familiar music to navigate through this challenging experience. One patient described the frightening experience of sedation. *'The world and your, everything else you know is kinda flipped upside down. . .it can be quite kinda scary. . .I seemed to be in my purgatory a little bit. . .'.* But music helped him to move through it safely. *'I really wanted to be able to put some music on. . .to push me into. . . exiting . . .that space. . . in a positive way. . . - rather than being scared.' (P12).* Others, who experienced hallucinations and delirium, used music listening to make sense of their surroundings, and ground their perceptions in reality. One patient described their hallucinations as *'less jarring'* when music was playing.

One patient described how music listening preserved his cognition:*' I have a clear mind, I don't know if that's due to the music or not, but . . .when I first wake up. . .I am confused a little*

*bit but. . .at least I know what's going on. . .' (P5)*. These cognitive effects were noticed by family as well. One woman credited her father's cognitive improvement after a stroke to music listening. '*The difference that I saw in him, was his cognitive engagement. . . . seeing something. . . spark in his eyes . . . raising his eyebrows, like. . .he KNEW something. . . especially at that time when. . .sometimes it was hard to know whether he was there. . .'* (FP1)

This observation was echoed by many friends and families. For example, one person marveled at his friend's cognitive abilities after a TBI. '*I've been just blown away by his cognitive. . . function. He's- it's completely not noticeable! [It] seems crazy to me, that he. . .hit his head so hard and there was bleeding in his brain. . . that's just. . . blown my mind' (FP11)*. In another case, a mother was amazed by her adult son's preserved linguistic and cognitive agility, after witnessing his ability to construct careful arguments despite having a TBI. She attributed this to his having listened to music when he was recovering on the ventilator. '*Four days after a traumatic brain injury. . . that trait has. . .come through again. . ..he didn't lose that with this horrible accident. [It's] absolutely phenomenal. To hear him talk like this.'(FP14)*

## Humanizing the hospital experience

Patients, family members, and staff all saw the use of recorded music as a way to humanize the ICU experience by restoring indivuality, identity, personal history, control, and independence, and by redressing the mechanized environment. Music reinserted personhood into the clinical space. Playing recorded music for patients helped remind clinicians that patients were human beings, encouraging staff to see past the diagnoses and the mechanistic environment surrounding them. One nurse explained, '*. . . we get very disease focused, see trauma, see disease, this [music] shows people [for who they are]'*. One resident physician offered that hearing music in a patient's room '*humanizes him to me'*. And one patient recovering from critical poly trauma explained that, prior to the offer of music, all of the human contact he experienced had been disease and injury oriented, but when staff offered to play music, he finally felt they were seeing and treating him as a *person*. Similarly, playing recorded music for her father helped one woman assert '*that he wasn't just a patient, he was someone who. . . was HIM. . . and that they're all these pieces to him that are important' (FP1)*.

Patients described music selections as closely aligned with their identity. Despite residual deficits after a serious TBI, one young man was immediate in his selection of a preferred song, using it to get his '*groove back'*, to feel more like himself '*at base'*. He explained that '*music has a. . .. property that is.. . . very . . . specific to . . . your identity (P15)'*. Music was usually chosen by patients or families, but in one example, a group of clinicians of similar age and background to a young, comatose man suggested a playlist after looking at photographs of the patient brought in by his family. The song choices delighted the family, who felt the staff had seen their brother for who he was and identified with him. Later they explained that asserting identity through personalized music was a form of advocacy. *Hopefully, if he hears it he knows that. . . people are thinking of him. . . he knows that, '. . .maybe someone vouched for me"(FP9)*.

Through the memories attached to music, people were reminded of who they were before their hospitalizations.

> *When you ask me what kind of music do you like?- that gave me a chance to remember the good times. . . the relationship with the prints of my heart. . .yes, yes it's made me think. . . it made me relax . . .the memories. . .I appreciate- that you givin' me that. (P17)*

Others described the feeling of being 'brought back' to themselves with music after life-changing trauma. '*it definitely just brought me back to.. ME and MY ROOTS. . .how I believe and how I see MY music and my positivity.' (P7)*.

Not all memories invoked through music were positive, *it's like a fast track to all of the emotions that link to that music and the experiences that you have to those songs. So all of the good times and the bad times' (P12)*. These potent facets of identity were valued nonetheless. *'Familiar music is like an old friend. Sometimes they're good, sometimes they're bad. . . but whatever it is, it's yours.' (P5)*.

Alongside individual memories, people associated music with their family identity. Patients and families desribed their music choices as symbolic of their 'childhood', or memories associated with their parents, siblings, and grandparents. Others were reminded of their neighborhoods or places of origin. *'It was my mom, my dad and also my hood, like, in the streets. We would always have a speaker playing. . .you'd go outside with music, and go inside- my neighbor's playing music' (P8)*.

Threats to identity left one man feeling that death was near; for him, selecting music steeped in family history was a natural choice. *'I felt like I was on like my ancestral plan, like I felt like my body and spirit were in two different areas' (P7)*.

Facing life threatening events and cast into chaotic settings, patients used music to normalize their environment. Some asserted that music was a 'need'. *'I think. . . that no matter what you do. . . music it's something that's so important in your life that you always need' (P5)*. One family member explained to her son, *'this was just . . ..a horrendous event, and I think introducing music that you're familiar with, or even not, but just music. . .would bring more. . . normalacy to the whole environment. . . you needed that.' (FP14)*. Restoring some feeling of normalcy also helped people re-situate. One patient explained that his '*biggest challenge*' after the accident was that he didn't feel '*normal*'. He went on to describe how listening to music at night in the ICU gave him something to '*relate to*', and '*feel like I'm in my bed, in my home bed*' (P5).

Normalcy achieved through music helped people begin to heal from their injuries and emotional pain. *'I felt like I had to have a certain sound in order to be able to get over certain traumas or even get over this accident. I felt like music uplifted me. . .. music brought me back to me feeling normal.' (P8)*

Meanwhile, mechanistic sounds in the ICU were a constant reminder of the trauma experienced by this group; several described feeling re-traumatized by alarms and oppressed by the unatural sounds of humidifiers, pumps, and machines. One patient used music to distract himself from the tube feeding pump, which he found upsetting. *'All you can hear is the feeding tube. You can hear and you can feel that something is pumping into your stomach. It's just disgusting. You don't know what's been pumping into your body.' (P5)*

Loss of control was another threat to the humanity of patients in the ICU. Choosing and playing music softened this experience. One patient explained that listening to sounds he could control made him feel '*safer*'. Similarly, one woman hoped music would ease her husband's loss of independence in ICU. '*He has been in control his whole life and the frustrating thing is that he can not move around . . . can't talk. . . and that's where this music . . . comes in' (FP3)*. Some patients recalled bad memories of being restrained in the ICU. One man wished for music to help distract him from this loss of control.

> *That's a really really bad feeling. . . . when you first wake up from ICU and . . . you're tied down, you're confused- you're just there 24 hours a day- it's a very bad feeling. . ..it feels like- it's worse than prison. . . But, you know, something else like music to distract you would be a better option. . .instead of. . .getting restrained, and. . . just hating it (P5)*

One young man with a traumatic amputation described the total loss of control in the hospital, *'everyone is telling you what to do, everything is very controlled. The way we eat- the time*

*limits. . . everything is a schedule.'* But picking and playing his own music enabled him to assert himself.

> *Listening to my own music, on my own time, when I want. . .that's. . .something that I could do. I could do it!. . . I might not be able to walk, I might not be able to see. . .but I'm able to change the song. I'm able to choose a song. Music, makes you feel. . . like not every- I'm not being controlled by this aspect.* (P8)

Regaining some control through the use of music, also helped people re-establish their autonomy. '*I feel like music..helps me just remember myself that I'm still independent. I'm still my own person. I could still make my own decisions'. (P8)*

### Providing a source of connection

In addition to feeling dehumanized, patients in this cohort experienced fear, loneliness, and lack of social connectedness. Listening to music faciliated connection for patients deprived of social contact and interaction. As explained by one patient '*my grandfather used to say that human beings are social creatures, and we can communicate through music and I firmly believe that'* (P10). Listening to music gave patients a way to connect to their families when restrictive visiting policies kept them apart. '*My family can't be around me in the room sharing with the love. So I felt the love through the music because of that connection that I made with the music.' (P7)*. Listening to music invoked memories of life outside the hospital, including times when people listened to music together. This was expressed by several family members who hoped music would lessen the loneliness of hospitalization. One family member explained,

> *the idea of. . . your friend being alone in a hospital seems pretty horrible, so just the idea of any music in there-The idea of him just being in a hospital room by himself. . .alone. It sounds horrible. It's like very lonely too. . . fact that there was music definitely made it seem nicer (FP11).*

Several patients complained of the feeling of time standing still, a psychic doldrum, which exacerbated feelings of lonliness and separation. Listening to music helped to moderate this experience.

> *. . . some people don't have their families here and you're just looking at the time, you're just looking at clock all the time waiting and seeing time pass by and you feel alone. Yes music helps. Music helps because you don't feel alone (P2).*

For another man, missing his family, listening to music was '*kinda a little bit of a connection to home'* (P11). Others compared the benefits of music directly to those of a visitor, '*Having a visitor is the like music- let's your mind wander to someplace else. . . helps you. . . get through the day' (P10).* Listening to music was akin to '*hearing another voice'* for one young man, a form of socializing. He explained, '*the chord progressions speak to you'.*

Patients were in search of connection, something to '*grab hold of'* psychologically and cognitively. Listening to music provided a means for patients to connect with their environment and circumstances despite being intubated and unable to speak. Several patients suggested that identifying preferred music ought to be an intake question during all ICU admissions, '*because it <u>does</u> help you connect when you can't say nothing, your mind is still processing these items and and attaching music to your happy moments in life. . .' (P7)*

One man suggested that music could even serve as a form of connection to people in coma, *"they tell you to talk to them, touch them. Because they want that connection. So I think music can provide that connection for them' (P5).* Hearing familiar music also helped patients to feel cared for and connected to the clinicians: familiar music *'might help me think that you know like I'm belong. . .belong to this nurse. . . like not belong to somewhere else. . . like belong to. . .here'.* (P5).

Importantly, personally meaningful music also helped patients connect with themselves. Music connected patients to their belief systems, restoring a sense of purpose by reminding them of who they are. *'Cuz we coulda easily put some hip hop on, and just drowned the noise out. But for me it was more like, I wanted. . . something to connect with and to <u>empower</u> me [to] know [that] I'm ALIVE and being alive I felt closer to my grandfather, closer to my brother. . .' (P7)*

### Improving psychological wellbeing

Patients described the use of music listening to address several aspects of their psychological wellbeing during their ICU admissions including: taking their mind off their problems, managing intrusive thoughts, processing emotions, accessing resilience, and experiencing hope and pleasure.

Several patients requested music during periods of extreme distress (anxiety, breathlessness, frustration). For some, the choice of music was not important; some patients suffering intense agitation or anxiety asked for staff to play '*anything*', stating '*it don't matter*'. Patients who had been trying to climb out of bed for hours, relaxed almost immediately once music was started. One young man nodded vigorously when clinicians asked 'should we restart the music' after he had started to cough and sit up in bed despite high dose sedation. Later, some patients explained that listening to music gave them something other than their suffering to focus on. One patient said, *'I think I was looking for, like, a distraction? . . . so it's like something to pay attention to, other than my own pain or whatnot (P14).* This view was endorsed by a young father who had sustained a life threatening injury:

> *'Strapped to the bed, tubes everywhere- and it [music] just made me feel like nothin', none of that was important you know? The music was just playing and soothing me and it just made me not think about all my all my other problems I had right at that moment'. (P7)*

Most of the patients in this group suffered the progression of time. As time stood still, patients felt worse emotionally. *'Time was. . .dragging on. Pain was higher. I was. . . in a worse mood, more depressed' (P12).* Listening to music gave patients something else to focus on. One patient plagued with hallucinations, endured them by focusing on breathing, counting each of his breaths. He gave himself points in a . . . *'mental. . .game that I'm playing where I just get points by surviving'.* Listening to music gave him another way to distract himself from the unease, *I could just breathe and I wouldn't have to worry about having to. . .meticulously keep track of how many times I'd breathed. (P14).*

Unfamiliar environments and uncertainty about the future also left patients and families feeling scared. Listening to music helped many feel 'safe'. Several patients reported using music to manage intrusive thoughts. One patient explained that music *'put[s] people. . .in more of a meditative state which isn't a bad place to be rather than. . . flashing through whatever trauma they've been through'. (P12).* One patient used music to get relief from thoughts of the accident. For him, music '*took away the accident, . . .it took away my hundred percent of seeing everything back to back to back.*' He explained,

*'it relax my mind. Cuz' my whole mind . . .was thinking about the pain and my trauma. . . When that thing played it took my pain, it took my pain. . . 'cuz it shifted my whole position. . . from the accident. . ..So that helped me,.' (P17)*

Flashbacks interfered with sleep for many, but listening to music helped patients to rest. '*I went to sleep listening to the music . . .. even for a long time in my sleep I was listening' (P17).*

Several of the patients in this cohort experienced significant loss associated with their hospitalizations. Listening to music facilitated emotional processing of their experiences, and, for some, listening to music was a comfort during a time of grief. *'It's very. . . painful to know that my life is gonna change drastically. . .Its like music . . . it's giving me a hug.' (P8)*

Others described the use of music to stimulate tears. '*Like, when you listen to a sad sad music, like you would want to cry, the tears sometimes come by itself, right*?' Emotions conjured by music gave some patients a way to express their sadness. One patient explained after the fact how a specific song would have helped him when he was alone in the ICU. *"If you guys had played that . . .I would probably feel a lot better. . . . . . .it's a song that really touches your heart. And it let goes of a lot of anxiety, it let goes of a lot of stress (P5).'*

For those who were unable to talk about the challenges they were facing, listening to music was a way to start to process grief. Two friends described the first moments they spent together in the ICU, '*we just sat there and listened to music together. . . We didn't know what else to talk about. So, we just listened'* (P11). Others experienced a numbness after their injuries. One man described how hearing music facilitated feeling again, allowing him to re-engage with his psychological experience of trauma and to access his own resilience.

*I think it changed how I felt about myself in that moment. . . .when I was on the operating table and I came in the ER, I felt I gave up already, I didn't think I was going to come out of it . . .I still didn't process anything. . . it was still very much numb and blank and . . ..when I started to process my own thoughts and and my own feelings it was when the music came on because it was it was attached to me and it was what I wanted. (P7)*

In addition to quieting instrusive thoughts and comforting the griefstruck, listening to music gave people hope and enjoyment. Hearing music reassured patients that no matter the physical outcome of their trauma, they would still find pleasure in life. '*. . . as soon as you know that your ears are working, you know whatever physical thing you're going to be left in, you're gonna be at least, you know, enough to still enjoy what you love (P12).* Even in the ICU, patients described music as filling their '*hearts with happiness'.* Family members observed that music '*brightens up the space'*; they hoped that playing and listening to music would bring joy to their loved ones. One man explained that while his days in the ICU were '*the worst days of his [sic] life'*—he '*did remember. . . enjoying the music'*. Others endorsed that listening to music gave them something '*fun to think about'.* For some people, music brought '*inner peace'.*

*To hear something familiar gave me something to look forward to when I was already at the end of my chain. I had already accepted that I was dead and I didn't think I was going to wake up . . . but then I knew that there was ONE thing that was familiar . . . and I was just like YES, thank G-D, that's my song right there! (P7)*

## Resolving the problems of silence

Silence was seen as problematic by most patients and their families. Without sensory stimulus, families worried that their loved ones would not be able to connect to their environment and

would be stuck without a means to emerge from unconsciousness. One parent of a semi-coma-tose young man observed: *'he is just sitting here in a blank room, you can't just be in a blank room, that isn't healthy'- [there's] nothing to come back to'*. Silence was seen as a form of aban-donment to those experiencing coma. One patient explained:

> . . .*a lot of people in coma, people just left them there, you know? Like people just left them in silence. They shouldn't do that. Talk to them, touch them. . . let them feel. Even if they don't feel anything. . .play some music at least. Let them know that, hey we're calling you. . .We're calling you from here. . .you need to come back. (P5)*

Silence was also associated with restraint and described as something to endure. One patient wished that someone had played music for him *'instead of leaving [him] just lying down there. . . in silence'*. Adding, *'. . .my hands were restrained and I was lying in silence and that's not a good feeling.' (P5)*

Other family members observed heightened fear response in their loved ones in silent, quiet rooms. *'It was so silent that everytime someone bumped a table or something. . . [it was] startling. . .. His eyes would get wide. . .why is it so silent'*? And one patient used music to mod-erate the sudden sounds of doors opening and monitors beeping, each of which triggered intrusive thoughts of the trauma he endured. *'. . .it helped me come up from- a little noises at first- because I was thinking and thinking- why me, why me all that time' (P17).*

Silence was also associated with depressed mood and dampened hope. One pair of siblings noticed a more 'upbeat' staff and joyful ambiance when music was playing compared to the sterile mood experienced in a silent room. They wondered if this change in atmosphere might help encourage their brother to recover.

> . . .*It's just so . . .quiet here . . .[the music makes it]. . . seem a little more upbeat and not so quiet. . . [it] feels a little bit . . .lighter. . ..it is a very serious situation, but if people are more positive, and if their tone could be a little more positive maybe that will also help him. . .their tone seems a little bit more healthier so. . . he might feed off of that or he might feel it (FP9)*

People worried that their loved ones would feel disoriented in silence, adding that hearing music may help a person to re-situate.

## Discussion

This study provides important insight into the potential benefit of music listening in critical care by identifying six under described uses of music listening for ICU symptom management: restoring consciousness; maintaining cognition; humanizing the hospital experience; facilitat-ing a source of connection; improving psychological wellbeing; and resolving the problem of silence. Social and demographic patient and family information and music selections varied widely in this study; however, the perceptions of the benefits of recorded music listening were largely shared. Overall, listening to personally relevant recorded music helped patients to achieve psychological and cognitive homeostasis across a spectrum of arousal and emotional states. At times, music stimulated consciousness and awakening; at other times, music lulled an overactive mind. Listening to music facilitated emotional processing for people experienc-ing numbness, but also distracted those burdened by fear and flashbacks.

Memories associated with music served as a stimulus for consciousness and cognition, and also facilitated psychological wellbeing during ICU hospitalization. Music related memories contribute to autobiographical memory helping patients to make sense of their environment, and to construct a feeling of identity and continuity [24, 27]. In our study a socially, culturally

ethnically, and economically diverse group of participants used music to regain control, assert their identity, and reconnect with their environment. Additionally, playing and listening to personally selected recorded music gave people a chance to be seen and accepted as individuals by the health care, strengthening the relationships between patients and clinicians. Description of the phenomena listed above, adds to understandings that explain the mechanism of action of music listening for symptom management in critical care.

## Music listening for psychological symptoms

The results of this study indicate that the potential uses of music listening are much broader than have been previously described and that few patients described the use of music listening for management of pain or anxiety. We found that potential applications for music listening were wide ranging including important symptoms for which there are few treatments available, specifically *psychological* pain. Alone with their thoughts, many patients experienced hallucination, fear and flashbacks. While some patients agreed that music helped them to feel more 'relaxed,' the mechanism for this was cognitive, through distraction. Preferred music was able to hold their attention, replacing the negative thoughts that had previously overwhelmed them. Relief from a racing mind also enabled some to get sleep and may explain improved sleep experiences associated with music interventions in critically unwell adults [36]. In other instances, listening to music gave traumatized patients a tool to process their emotions, a critical step in managing PTSD [37] which is common among ICU survivors. Playing personally relevant music represents an opportunity for collaboration with psychosocial providers (e.g. music therapists, violent injury prevention professionals, social workers, chaplains and psychologists) to more deeply address grief and trauma-related needs.

Consistent with prior studies, patients in this group also experienced loneliness, de-humanization, dependence, and loss of control [2, 8, 38]. Listening to recorded music helped patients to endure these physical and psychological restraints. Feelings of powerlessness and loss of independence are threats to resilience and predictors of post intensive care syndrome [39]. Patients asserted control of their environment and their thoughts when making music selections that made them feel normal and capable. Additionally, being able to play personally relevant music helped patients and families to be 'seen' by clinicians and feel respected.

Despite research asserting the value of family presence to prevent delirium [40], many visitation policies limit patients' contact with their families, leaving patients few options to engage in meaningful social contact. In our study, patients summoned their social lives when listening to music through reminders of their family members, ancestors, social events, and day to day habits. Many ICU survivors suffer from depressed mood, and sadness is a commonly reported symptom during ICU hospitalization. But listening to recorded music may mitigate emotional pain and augment positive feelings such as pleasure, joy, and hope, and it may also facilitate coping. Patients in our study reported that when listening to music they were reminded of happy times and felt motivated to survive. While there are no validated measures for many of these symptoms, aggregate instruments such as the Intensive Care Psychological Assessment Tool (IPAT) [41] and the Profile of Mood States (POMS) [42] may serve as proxy measures for future studies of MBI designed to prevent and treat psychological distress in critical care.

## Music listening for cognitive stimulation

Prior works exploring the use of music in critical care have examined the use of music listening as a sedative. Rather than sedate, music listening helped unconscious participants in this group to wake up. This phenomenon aligns with recent research demonstrating improved level of consciousness after music intervention among people experiencing coma [43, 44] and TBI

[45]. We believe that this is the first study to describe the subjective value of consciousness for patients experiencing ICU hospitalization. In our study, people fought to regain consciousness, preferring an awake state to a sedated one where they had little control over their thoughts, bodies, and environment. Once awake and conscious, patients used music to process their trauma, to situate, to take control and to connect.

Findings from this study also contribute to research that explains the mechanism of action of music listening for symptom management in critical care. Patients and families in our study described music as a 'trigger,' a recognizable stimulus to encourage their consciousness. This is consistent with studies that show that listening to music engages a bi-hemispheric network related to attention, semantic processing, memory, sensori-motor and emotion [24, 27]. For example, the dorsal medial pre-frontal cortex is activated while listening to familiar songs from the past. This phenomenon has also been observed in comatose adults who showed increased evidence of semantic recognition on functional MRI after listening to familiar music [46].

Participants also attributed patients' cognitive agility to the use of music listening in our study. Listening to music, even during coma, may enhance cognitive recovery by keeping brain cells 'in use.' a critical component of neurogenesis [47]. Data demonstrate that the plasticity of the central nervous system may benefit from a rich stimulation regimen [48, 49]. In fact, improved cognitive functioning has been associated with the use of music and music therapy after brain injury [50, 51].

In order to be of benefit for neuro cognitive rehabilitation, some argue that sensory stimulation must be organized, (e.g. music), and caution against sensory bombardment (e.g. alarm, mechanistic sounds) [52]. While much has been written about the risks of noise in ICU, few have examined the risks of silence, isolation, and sensory deprivation in critical care. Sensory deprivation is a known risk factor for delirium in ICU [53] and is also associated with diminished brain plasticity [54]. Sensory deprivation is also associated with isolation after TBI and may compound the negative effects of injury [54]. Current clinical models of critical care favor sensory regulation limits, but purposeful sensory stimulation may be better for brain health [48]. Listening to music, even during coma, may be enough to engage a mind otherwise threatened by lack of stimulation, ultimately preventing delirium. Indeed, current best practice guidelines for music listening during recovery from disordered consciousness favor a mix of salient music listening with periods of rest [55]. Patients in our group described feelings of abandonment and restraint associated with silence, exacerbating the stress of their hospitalizations. Here too, playing music moderated these experiences. Patients used music to pass time and connect with their identity through memories invoked by the music while trapped in an isolated, mechanistic environment.

## Limitations

There are several limitations to this study. First, we don't know how often or when (times of day) patients listened to music or if preferred music changed alongside shifts in clinical conditions. Nor do we know how well aligned the family-chosen music was with patient choice once able to engage. In fact, the reliability of surrogate selection of personally relevant music for unconscious people has not been established and represents an important area for future research. Another limitation is that all patients in this study identified as men. Interviewing a more gender diverse group may confer new insights into symptom experience and the use of music during critical illness. Notably, several of the other informants in this study were women and their views were consistent with the participants we interviewed. Similarly, all patients in this study were admitted to a single, urban trauma center. Since the culture of

critical care units varies, including other centers would add to the understandings developed here. All interviews were conducted by a single researcher, whose preconceived views of critical care may have influenced the line of questioning. This was mitigated through reflection and discussion with other members of the team throughout the study period. Finally, this study was conducted during a period of severely restricted visitation and societal stress related to the Covid-19 pandemic which may have heightened the experience of isolation and loneliness described by the respondents. However, both problems have long been reported in critical care and the insights gained through this research may be helpful to patients whose loved ones are not able to be with them for other reasons.

## Conclusion

This novel study describes the use of personalized recorded music listening to manage the psychological symptom experience of ICU hospitalization. Our results suggest that personalized music may have an effect on cognition, confusion, intrusive thoughts, sleep, fear, loneliness, and helplessness. Participants used music listening to take control of their thoughts, their bodies, and their environment. While the symptom experience was diverse in this group, common amongst all was a desire to achieve cognitive and psychological homeostasis. These findings add to the growing understandings of the mechanism of action of music as a meaningful cognitive stimulus and help to develop a framework for understanding how music works to help people recover from critical illness. Delivery of recorded music is an equitable, patient-centered intervention that can easily be tailored to individual needs. Given the prevalence of long-term cognitive and psychological morbidity experienced by ICU survivors, personalized MBIs designed to treat psychological distress are likely to be of great benefit.

## Supporting information

**S1 File. Interview guide.**
(DOCX)

**S2 File. Other informants.**
(DOCX)

## Acknowledgments

The authors wish to acknowledge the contribution of Michael Texada, for his early assistance with the original concept for this research and for his insight during synthesis of the findings. As well, we acknowledge the patients, friends and family members who provided their perspectives on the use of music listening and who shared deeply personal stories related to their hospital experiences.

## Author Contributions

**Conceptualization:** Rebecca Menza, Jill Howie-Esquivel, Julin Tang, Heather Leutwyler.

**Data curation:** Rebecca Menza, Tasce Bongiovanni, Heather Leutwyler.

**Formal analysis:** Rebecca Menza, Jill Howie-Esquivel, Tasce Bongiovanni, Heather Leutwyler.

**Investigation:** Rebecca Menza.

**Methodology:** Rebecca Menza, Jill Howie-Esquivel, Julene K. Johnson, Heather Leutwyler.

**Project administration:** Rebecca Menza.

**Supervision:** Julene K. Johnson.

**Validation:** Jill Howie-Esquivel, Heather Leutwyler.

**Writing – original draft:** Rebecca Menza.

**Writing – review & editing:** Jill Howie-Esquivel, Tasce Bongiovanni, Julin Tang, Julene K. Johnson, Heather Leutwyler.

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
