## [Decision Letter · Decision Letter 0]

18 Jul 2024

PONE-D-24-19043Personalized music for cognitive and psychological symptom management in critical care: A qualitative analysis of patient experiences during mechanical ventilationPLOS ONE

Dear Dr. Menza,

Thank you for submitting your manuscript to PLOS ONE. After careful consideration, we feel that it has merit but does not fully meet PLOS ONE’s publication criteria as it currently stands. Therefore, we invite you to submit a revised version of the manuscript that addresses the points raised during the review process.

Your research reveals exciting insights of icu patients listening to their favorite music. However, there is the need for a minor revision from the reviewers and I kindly ask to address all points of criticism.    

We look forward to receiving your revised manuscript.

Kind regards,

Alexander Wolf

Academic Editor

PLOS ONE

2. In the online submission form you indicate that your data is not available for proprietary reasons and have provided a contact point for accessing this data. Please note that your current contact point is a co-author on this manuscript. According to our Data Policy, the contact point must not be an author on the manuscript and must be an institutional contact, ideally not an individual. Please revise your data statement to a non-author institutional point of contact, such as a data access or ethics committee, and send this to us via return email. Please also include contact information for the third party organization, and please include the full citation of where the data can be found.

Additional Editor Comments (if provided):

Reviewers' comments:

Reviewer's Responses to Questions

**Comments to the Author**

1. Is the manuscript technically sound, and do the data support the conclusions?

Reviewer #1: Yes

Reviewer #2: Yes

2. Has the statistical analysis been performed appropriately and rigorously? 

Reviewer #1: Yes

Reviewer #2: I Don't Know

3. Have the authors made all data underlying the findings in their manuscript fully available?

Reviewer #1: Yes

Reviewer #2: No

4. Is the manuscript presented in an intelligible fashion and written in standard English?

Reviewer #1: Yes

Reviewer #2: No

5. Review Comments to the Author

Reviewer #1: (A version of this statement is also included in the attached review comments).

Many thanks to this team for your extremely resonant, meaningful contribution to the literature surrounding wide-ranging impact of music listening for patients and their families in ICU contexts. I particularly appreciate the way your work emphasizes patient perspectives and lived experiences, which both amplifies the patient’s voice in meaningful ways while simultaneously developing much-needed theory surrounding mechanisms of MBI application in the ICU.

The interview quotes you selected were evocative, especially those discussing the complexities of being hospitalized (e.g., grief, restraint, dehumanization), and I thought these were very thoughtfully positioned in tandem with reflections about how music listening impacts patients and their family members across many domains. I also applaud your willingness to push back against the status quo in terms of the types of symptoms that are “typically” studied as outcome measures in research surrounding MBI (e.g., pain, anxiety), allowing patients and their families to take control of the narrative and provide direction for future research.

On a personal note, the impact of music described as described by the patients in this project more fully aligns with my experiences of sharing music with patients in ICUs than those studies which focus on pain and anxiety. These patient perspectives speak to the nuance of music as a holistic, embodied experience, and as something reaches beyond symptom mitigation alone. I appreciate the ways in which your work sits in this space of holding music as a both/and experience.

Your manuscript was written accessibly, with clear descriptions of your grounded theory data analysis and interpretation processes.

More comments with a higher level of specificity and curiosity surrounding elements of your project are included in the attached reviewer comments. In particular, I would draw attention to clarity of language (patients versus patients and families; symptoms), clearer descriptions of what is meant by diversity, and more nuance surrounding the both/and of music implementation in neuro ICUs.

Reviewer #2: Congratulate the authors for such magnificent work, integrating a qualitative methodology into the design of their study.

In the methodology section: the introduction of the method contains information that should be in the previous section. Damasio, Nightingale...These are data from the introduction and not from the methodology section.

In the methodology section: with what scales did you measure the level of consciousness? When they say devices near the bed, in unconscious patients? (it is not clear to me). What happens with external noise such as mechanical ventilator alarms, infusion pumps, and other confounding variables such as aspiration of secretions, tolerance to orotracheal intubation? Did the patients wake up with the music playing? Or did they only have it while they were asleep/awake?

In the results section: Excellent presentation of the qualitative results.

But regarding the quantitative results, I have lacked the ability to know times (unconscious, conscious patients), there were no limits on time or times, but how long were they listening to music, how many times?, according to each patient? What levels of consciousness did they have depending on the moment they were listening to the music? There are 14 patients. I think they could improve table 2, showing this data.

6. PLOS authors have the option to publish the peer review history of their article (what does this mean?). If published, this will include your full peer review and any attached files.

Reviewer #1: No

Reviewer #2: **Yes: **Marina Mateu-Capell

---

## [Author Response · Author response to Decision Letter 0]

1 Sep 2024

Dear Dr. Wolf and Reviewers, 

Thank you for the supportive and helpful review of our paper: Personalized music for cognitive and psychological symptom management during mechanical ventilation in critical care: A qualitative analysis. It is wonderful to hear that the stories and results resonated with the reviewers. We found it difficult to choose from the quotes as the participants were incredibly forthcoming in rich and detailed ways. We appreciate the careful reads you have given this paper and the suggestions you have made that will improve this manuscript. Please find attached our responses to your feedback including clarifications, explanations and a summary of changes. Also please note we have changed the title (slightly) to address an area of concern. 

2. Has the statistical analysis been performed appropriately and rigorously?

Reviewer #1: Yes

Reviewer #2: I Don't Know

RESPONSE: This is a qualitative study only; statistical analyses are not typically a part of qualitative studies. Therefore, we do not include a statistical analysis for the data presented in this paper. 

3. Have the authors made all data underlying the findings in their manuscript fully available?

Reviewer #1: Yes

Reviewer #2: No

RESPONSE: There are restrictions on publicly sharing the human research participant data for this study due to participant privacy outlined in our IRB (we do not have permission to share the de-identified data in a repository). Further, the small number of deeply personal interviews contain highly sensitive material, collected from vulnerable people, during a specific period of time and cannot be shared publicly due to risks that participants may be able to be identified through indirect means. In order to maintain confidentiality and compliance with our IRB and to meet our ethical obligations and the data sharing requirements for publication in PLOS, we are amending our Data Statement to include a third-party (tbd) who can manage data requests on a case-by-case basis on our behalf. We have reached out to our IRB and institutional data management leadership to identify an appropriate agent. In the meantime, the authors are happy to field requests as they may arise. 

4. Is the manuscript presented in an intelligible fashion and written in standard English?

Reviewer #1: Yes

Reviewer #2: No

RESPONSE: All authors reviewed the manuscript for any unclear, incorrect, or ambiguous language, and we did not identify any sentences that needed improvement. Please identify specific instances, and we are open to making the manuscript more intelligible. 

REVIWER: In the methodology section: the introduction of the method contains information that should be in the previous section. Damasio, Nightingale...These are data from the introduction and not from the methodology section.

RESPONSE: We moved the description of the theoretical frameworks into a subsection of the Introduction, under the subheading ‘Theoretical Framework’.

REVIEWER: In the methodology section: with what scales did you measure the level of consciousness? When they say devices near the bed, in unconscious patients? (it is not clear to me). What happens with external noise such as mechanical ventilator alarms, infusion pumps, and other confounding variables such as aspiration of secretions, tolerance to orotracheal intubation? Did the patients wake up with the music playing? Or did they only have it while they were asleep/awake?

RESPONSE: This was a qualitative study only of the views of patients and families who listened to personally relevant music while mechanically ventilated at some point during their ICU stay. 

We did not collect measures of level of consciousness during this study. Patients were of varied and changing levels of consciousness during their ICU stay for a variety of reasons (TBI, sedation). Similarly, patients listened to music during a variety of states of consciousness throughout their ICU stay.

In general, speakers and tablets were placed near (within </= 5 feet) for un/semi-conscious patients. After people regained consciousness, we offered them a choice of headphones or speakers/devices. This is described in the section ‘Recorded music listening.’

Because this was a qualitative analysis of an uncontrolled music listening experience, external noises (suctioning, aspiration, etc) were not viewed as confounding variables. Rather, these were all part of the overall experience of being in the ICU on a ventilator. 

Because we did not observe all music listening experiences we cannot report on the number of patients who were listening to music while asleep, sedated or in a coma state, nor can we report the number of patients who woke up while listening to music, or who listened to music once awake. 

REVIEWER: In the results section: Excellent presentation of the qualitative results.

But regarding the quantitative results, I have lacked the ability to know times 

they listening to music, how many times?, according to each patient? 

RESPONSE: This was a qualitative study only. (This was not the qualitative of a quantitative study). For this analysis we did not perform any measurements, instead we report the views of patients and their families, gathered through interviews and conversations during and after ICU hospitalization.

We did not use a standardized protocol for music listening in this study. Our study focuses on querying the experiences of music listening after the fact. To make this clearer, we removed the words ‘use of’ from the header which now reads Recorded Music Listening

We did not collect information about how often or when patients listened to music or how the listening practices changed during the ICU stay. We added these points as limitations to our study. 

In the limitations: First, we do not know how often or when (times of day) patients listened to music or if preferred music changed alongside shifts in clinical conditions. 

REVIEWER: What levels of consciousness did they have depending on the moment they were listening to the music? There are 14 patients. I think they could improve table 2, showing this data.

RESPONSE: We did not measure level of consciousness during this study. As we describe above, patients experienced a range of levels of consciousness throughout the ICU hospitalization and listened to music during a spectrum of states of consciousness. We are reporting the patient and family descriptions of these experiences in this paper. 

We conducted 14 interviews for this analysis, but the data used to develop the themes also includes notes from several other people (patients, families, clinicians) who used music during the hospitalization. We made reference to Appendix 2 in the original submission, but neglected to include it. Some of the other informants are now listed in Appendix 2. 

Appendix 2: Other Informants

Parent of young man with coma after a motor vehicle crash (MVC)

Parent of young man with coma after pedestrian versus automobile (PVA)

Partner of young man with traumatic brain injury (TBI) after bicycle crash

Sibling of middle-aged man with cerebral vascular accident (CVA) and coma after MVC

Middle aged man with blunt chest trauma

Bedside registered nurses 

Sibling of young woman with TBI after MVC

Medical doctors of neurology, critical care, surgery and neurosurgery: professors, and trainees 

Spouse and sister-in-law of man with TBI 

Middle aged woman after abdominal surgery

Middle aged man with abdominal injuries 

Family of older woman after CVA

Advanced Practice Providers on the trauma service

Uncle and parent of young man with assault related injuries

Parents and sibling of young man with assault related injuries and TBI

Siblings of young man with assault related injuries

REVIEWER: Introduction Definition of MBI: I noted no mention of music therapy within the manuscript, and wonder if it might make sense to add parentheses following discussion of specialized training (music therapy) as one such example of specialized training related to MBI delivery. 

RESPONSE: We amended the language in the text to be:

When music is used in the clinical setting to achieve a health-related goal such as symptom management, it is called a music-based intervention (MBI) [12]. MBIs may be delivered by a credentialled music therapist with specialized training (music therapy) or by a health care provider (music medicine) [13, 14]. 

REVIEWER: Limitations of previous research: I want to thank the authors for citing limitations of previous research (especially lack of participant diversity which therein results in more diverse music selection), and in particular, your early emphasis on the importance of personally salient music choices rather than provider-selected or standardized music. However, I do feel some additional context may be needed re: why previous studies have used less stimulating music (e.g., slower, instrumental), particularly in a neuro ICU context. This feels relevant to me given the constant risk of overstimulation and lack of ability for an unconscious patient to consent or verbalize felt experiences while music is played. As written, this statement appears dismissive of these choices to control music selection when there are, in reality, complex clinical and sensory factors at play. 

RESPONSE: The practice in our ICU (including in the neuroscience ICU, guided by the medical directors of neurosurgery and neurocritical care) has been to encourage the most familiar forms of cognitive stimulation with an overall aim to try to awaken people from unconsciousness/coma, and to observe patients carefully for their tolerance of this (vital signs, facial expressions). This fits with movements in the critical care community to avoid sedation and to encourage awake ICUs. Saying that, we acknowledge the lack of evidence exploring the use of uncontrolled music selection, and the value or safety of stimulating music for unconscious/comatose patients. The topic of controlled music selection (especially as it relates to tempo, mood, language) merits further consideration and is an area of interest for our group for future research.

We removed the language: “to slow tempo instrumental pieces” to avoid oversimplifying the matter, and we have changed the words “ignores” to “may contravene” to be less dismissive. 

REVIEWER: Diversity: I will discuss this in greater detail below, but wonder if you might specify what you mean by diversity when referencing previous studies.

RESPONSE: We added ‘socio-cultural’ in this portion of the text to be concise and are more detailed later in the methods:

“Lack of participant socio-cultural diversity is another important limitation in North American studies of MBI in MV adults”.

REVIEWER: Methods I found it interesting that the convenience sample included controlled music listening (e.g., folx who participated in your pilot MBI study) AND uncontrolled music listening as reported by physicians/staff who observed music listening taking place. I assume these are two very different kinds of music listening experiences (e.g., participants in a research project versus participants self-motivated to listen to music of their own accord) and wonder how, if at all, this impacts your findings.

RESPONSE: We may have created some confusion through the use of the words ‘pilot study’. The ‘pilot’ was simply to develop and implement a music listening program and see how people responded to it; there were no ‘controlled music listening’ sessions for this study. Most, but not all, of the patients listened to music chosen by their families when they were un/semi-conscious or sedated, and chose their own music as they regained consciousness. As you point out, there is likely a real difference in the music listening experience when a person is semi/unconscious (akin to a controlled listening experience) and when it is patient-initiated. We removed reference to the pilot study to clarify this and will address these differences more carefully in the limitations/discussion. 

REVIEWER: You mention a diverse sample again without clear description/definition of your team’s understanding of a diverse sample (e.g., Diversity of race? Cultural location? Age? Medical status/disability? Gender identity? Music listening preferences? Etc). While I acknowledge that you describe this somewhat in your description of study demographics in the results, I believe a more clear definition, especially at first mention of lack of participant diversity as a prior study limitation, would be meaningful and further contextualize the novel nature of your work.

RESPONSE: We added language to clarify what is intended for a ‘diverse’ sample. See below:

Purposive sampling was used to ensure a socially, culturally, ethnically, linguistically, and economically diverse sample of respondents and was ongoing until thematic saturation was achieved.

REVIEWER: Results Because patients AND family members participated in this project, I’m curious how your team grappled with the differences in these lived experiences during coding – particularly given that if only a family member was interviewed, they cannot actually speak to the patient’s lived experience (only their lived experience of witnessing what the patient is experiencing). This was not cited as a limitation later in your manuscript and is also not captured in your article title (“A qualitative analysis of patient experiences”). I suggest amending the title to “patient and family experiences” to be more descriptive of your participant group, and to consider adding a small portion of discussion surrounding the benefits and limitations of equating patient and family experiences as evidence of the same lived experience – as, in my opinion, they are not. 

RESPONSE: Thank you for raising this important question. We thought about this a lot and acknowledge that these perspectives are not the same. As you noted there were 3 interviews of family members only, and the other 11 were either of patients alone, or of patients with their family members present. We wanted to make sure that patients who could not [ever] speak for themselves still had a voice and that some part of their experience was incorporated into the analysis. Including their physical gestures was one way, but this too involved a certain level of interpretation (often by clinicians). Rather than ‘trust’ ourselves, we thought it better to look to families to help us understand what they felt their loved one was going through. We were [somewhat] assured that these views were a reliable representation after reflecting on data from prior studies that suggests that family/friend assessment of symptoms are highly correlated with patients’ self-report (https://pubmed.ncbi.nlm.nih.gov/22890258/). 

We decided not to call these ‘family-experiences’ of music listening since they were views of families of the patient’s experiences (not their own experiences of the music listening program). 

As an extra layer of caution, we coded all family-member perspectives for their views of the patients’ experiences and used these codes to confirm and illustrate concepts identified in the patient interviews. We added some language to the methods section to make this clearer and also some to the discussion. 

‘Family-member perspectives were coded for their views of the patients’ experience and were used to confirm and illustrate themes and concepts identified in the patient interviews.’

Regarding the title of the manuscript. We acknowledge that some of our findings are indeed adjacent to a ‘lived experience’. However, because we used a Grounded Theory methodology (and not Phenomenology) to frame our interviews, data collection, coding and analysis, we have positioned this paper around the social process about what is occurring: the why, what for, how, about personally relevant music listening. We hav

---

## [Decision Letter · Decision Letter 1]

3 Oct 2024

Personalized music for cognitive and psychological symptom management during mechanical ventilation in critical care: A qualitative analysis.

PONE-D-24-19043R1

Dear Dr. Menza,

We’re pleased to inform you that your manuscript has been judged scientifically suitable for publication and will be formally accepted for publication once it meets all outstanding technical requirements.

Kind regards,

Alexander Wolf

Academic Editor

PLOS ONE

Additional Editor Comments (optional):

Reviewers' comments:

Reviewer's Responses to Questions

**Comments to the Author**

1. If the authors have adequately addressed your comments raised in a previous round of review and you feel that this manuscript is now acceptable for publication, you may indicate that here to bypass the “Comments to the Author” section, enter your conflict of interest statement in the “Confidential to Editor” section, and submit your "Accept" recommendation.

Reviewer #1: All comments have been addressed

Reviewer #2: All comments have been addressed

2. Is the manuscript technically sound, and do the data support the conclusions?

Reviewer #1: Yes

Reviewer #2: No

3. Has the statistical analysis been performed appropriately and rigorously? 

Reviewer #1: N/A

Reviewer #2: N/A

4. Have the authors made all data underlying the findings in their manuscript fully available?

Reviewer #1: No

Reviewer #2: Yes

5. Is the manuscript presented in an intelligible fashion and written in standard English?

Reviewer #1: Yes

Reviewer #2: Yes

6. Review Comments to the Author

Reviewer #1: I feel the author team substantially addressed my comments and questions from the original review and believe the manuscript is now ready for submission. The manuscript is methodologically sound and a novel contribution to the literature. Statistical analysis is not relevant for a qualitative project of this nature, and making full interview transcripts public could pose a risk to confidentiality for participants. The manuscript is well written and clear. I have uploaded additional comments in a separate document, but will also include below.

R1: Personalized music for cognitive and psychological symptom management during mechanical ventilation in critical care: A qualitative analysis.

General comments:

Thank you to the author team for your comprehensive reflections and responses to our queries. I appreciate all of the changes you made to this manuscript, and find it improves clarity in terms of language choice(s), differentiation between the pilot project and qualitative arm, and differentiating patient vs family perceptions about music listening in the grounded theory analysis.

To summarize, I appreciated the following changes which include:

• Title modification for clarity

• Clarification that MBIs can be delivered by both music thearpists and healthcare providers

• Clarification of socio-cultural identities in reference to participant diversities

• Clarification of the connection between the two projects (uncontrolled music listening pilot + qualitative analysis of patient and family perceptions)

• Addition of clarifying statement re: coding of family member perspectives to further amplify emergent themes from patient interviews

• Careful labeling of patient and family viewpoints

• Title modification to table 3 to capture nuance of symptoms vs experiences

• Addition of language surrounding potentials of music to evoke strong feelings to include opportunities for collaboration with mental health providers on the team

• Emphasizing best practice guidelines for music listening and DOC ( salient music + periods of rest)

• Amending language of interview guide for clarity

I hope your group will continue to explore gaps surrounding use of music for patients with DoC/in ICU, particularly the effects of different aspects of salient music (e.g., stimulating versus slow/calming), impact of surrogate selection of music when patients are unable to self-select, and further explorations of lived experiences of music listening in critical care. It would also be wonderful to better capture more robust clinical and music-listening specific data in future music listening projects, including clinical status of patients (e.g., GCS or CRS ratings at time of music listening) and inviting patients/families/staff to maintain logs of music listening frequency/length/time of day as well as the songs/genres listened to.

Thank you for your excellent contribution and amplification of patient and family experiences!

Reviewer #2: As a qualitative study, it still has its strengths in the opinion of relatives and ICU patients. It is a pity that this opportunity was not taken to collect data on technical aspects, such as those mentioned: type of music, times per day used, etc. My recommendation for future research is to consider this data, as well as the hypnotic level of patients while listening to music, as well as the confounding variables discussed, especially on the subject of the use of sedatives.

7. PLOS authors have the option to publish the peer review history of their article (what does this mean?). If published, this will include your full peer review and any attached files.

Reviewer #1: No

Reviewer #2: No

---

## [Editor Report · Acceptance letter]

14 Oct 2024

PONE-D-24-19043R1 

PLOS ONE

Dear Dr. Menza, 

I'm pleased to inform you that your manuscript has been deemed suitable for publication in PLOS ONE. Congratulations! Your manuscript is now being handed over to our production team.

Kind regards, 

on behalf of

Dr. Alexander Wolf 

Academic Editor

PLOS ONE